

# Similarities and differences in the microbial structure of surface soils of different vegetation types

Yong Jiang[1,*], Wenxu Zhu[2,*], Keye Zhu[2], Yang Ge[2], Wuzheng Li[3] and Nanyan Liao[3]

[1] Key Laboratory of Ecology of Rare and Endangered Species and Environmental Protection, Guangxi Normal University, Ministry of Education, Guilin, China

[2] College of Forestry, Shenyang Agricultual University, Shenyang, China

[3] Guangxi Fangcheng Golden Camellias National Nature Reserve, Fangchenggang, China

[*] These authors contributed equally to this work.

Corresponding author
Nanyan Liao, lny605605@sina.com

## ABSTRACT

**Background**. Soil microbial community diversity serves as a highly sensitive indicator for assessing the response of terrestrial ecosystems to various changes, and it holds significant ecological relevance in terms of indicating ecological alterations. At the global scale, vegetation type acts as a major driving force behind the diversity of soil microbial communities, encompassing both bacterial and fungal components. Modifications in vegetation type not only induce transformations in the visual appearance of land, but also influence the soil ecosystem's material cycle and energy flow, resulting in substantial impacts on the composition and performance of soil microbes.

**Methods**. In order to examine the disparities in the structure and diversity of soil microbial communities across distinct vegetation types, we opted to utilize sample plots representing four specific vegetation types. These included a woodland with the dominant tree species *Drypetes perreticulata*, a woodland with the dominant tree species *Horsfieldia hainanensis*, a *Zea mays* farmland and a *Citrus reticulat* a fields. Through the application of high-throughput sequencing, the 16S V3_V4 region of soil bacteria and the ITS region of fungi were sequenced in this experiment. Subsequently, a comparative analysis was conducted to explore and assess the structure and dissimilarities of soil bacterial and fungal communities of the four vegetation types were analyzed comparatively.

**Results**. Our findings indicated that woodland soil exhibit a higher richness of microbial diversity compared to farmland soils. There were significant differences between woodland and farmland soil microbial community composition. However, all four dominant phyla of soil fungi were Ascomycota across the four vegetation types, but the bacterial dominant phyla were different in the two-farmland soil microbial communities with the highest similarity. Furthermore, we established a significant correlation between the nutrient content of different vegetation types and the relative abundance of soil microorganisms at both phyla and genus levels. This experiment serves as a crucial step towards unraveling the intricate relationships between plants, soil microbes, and soil, as well as understanding the underlying driving mechanism.

## INTRODUCTION

Soil is a vital resource that sustains the livelihoods of the global population, impacts various ecosystem functions, and directly and indirectly affects human health and well-being (*Bach et al., 2020*; *Lehmann et al., 2020*). Among the soil components, microorganisms play a crucial role in nearly all ecological processes and exhibit the highest abundance, diversity and metabolic activity, serving as a critical "link" that maintains ecosystem services (*Rampelotto et al., 2013*; *Mendes et al., 2015*; *Wang et al., 2019a*; *Wang et al., 2019b*). They are actively involved in the material cycle and energy transformation within ecosystems (*Van der Heijden, Bardgett & Van Straalen, 2008*; *Cardinale et al., 2011*; *Jing et al., 2015*; *Delgado-Baquerizo et al., 2016*). The major groups comprising soil microorganisms include mainly archaea, bacteria, fungi and protozoa (*Bach et al., 2020*; *Fierer, Wood & Bueno de Mesquita, 2021*). The diversity of soil microorganisms serves as a sensitive indicator of the changes in the terrestrial ecosystem and holds significant ecological significance in assessing alterations in the ecological environment (*Bardgett & Van der Purren, 2014*; *Chen et al., 2020*; *Schloter et al., 2018*). Studies have demonstrated that the loss of soil microbial diversity and simplification of soil microbial community composition can potentially compromise multiple ecological services, such as plant diversity, litter decomposition, nutrient utilization and nutrient cycling, thereby posing threats to ecosystem sustainability (*Bardgett & Van der Purren, 2014*; *Bahram et al., 2018*). Therefore, it is crucial to investigate the composition and diversity of soil microbial communities under different vegetation types, particularly in the context of China's extensive vegetation construction and the complexity of vegetation cover types.

Biodiversity plays a crucial role in maintaining the proper functioning of ecosystems, and the conversion of land for human use has resulted in a substantial reduction of biodiversity in primary habitats, estimated to be 13.6% (*Newbold, 2018*). Human activities have had a significant impact on terrestrial ecosystems, and it is projected that global biodiversity will decrease by 3.4% by the end of the 21st century, which will have a s detrimental effect on ecosystem function in many parts of the terrestrial biosphere (*Newbold et al., 2015*). Given the importance of biodiversity in ecosystem function, there has been a growing body of research focusing on biodiversity and community structure at local scales in recent years (*Newbold, 2018*). Soil microbes power all biogeochemical cycles on Earth and are an important basis for ecosystem function, influencing the planet's biodiversity (*Loreau et al., 2001*; *Santillan, Constancias & Wuertz, 2020*).

Land use change is a significant environmental factor that can profoundly impact soil environmental factors, nutrient conditions and biological interactions (*Engelhardt et al., 2018*; *Wang et al., 2019a*; *Wang et al., 2019b*; *Fang et al., 2020*). Consequently, it has a substantial influence on soil microbial community diversity and building processes (*Cheng et al., 2021*). It has been shown that land use type explained 97% of the variability in soil quality indices and that different vegetation measures had significant effects on vegetation composition and structure, biomass, litter, soil moisture, soil nutrients and soil microorganisms of the ecosystem (*Xu et al., 2014*; *Zhang et al., 2011*; *Liu et al., 2022a*; *Liu et al., 2022b*). Changes in vegetation type not only alter the landscape appearance of the land,

but also the material cycle and energy flow of the soil ecosystem, while having a profound impact on the structure and function of soil microorganisms (*Wan & He, 2020*; *Zhang et al., 2013*; *Tian et al., 2017*; *Delgado-Baquerizo et al., 2018*). Moreover, modifications in plant community composition can indirectly influence microbial diversity and activity by altering the input of carbon resources into the soil through the production of various apoplastic and root secretions (*Zhong et al., 2020*).

Soil microorganisms play a crucial role in the interaction with plants. Numerous studies have demonstrated that vegetation type is a key driver of soil microbial diversity, encompassing both bacterial and fungal communities, on a global scale (*Delgado-Baquerizo & Eldridge, 2019*; *Chu et al., 2020*). In temperate forests, plant diversity has been identified as a significant determinant of subsurface soil microbial community composition (*Prober et al., 2015*). Soil microorganisms are highly sensitive to environmental changes, and discrepancies in dominant species, microenvironmental improvement, material metabolism, and disturbance history between vegetation types can considerably shape the evolution of soil microbial communities (*Ayres et al., 2009*; *Wan & He, 2020*; *Zhang et al., 2013*; *Tian et al., 2017*; *Delgado-Baquerizo et al., 2018*). Above-ground vegetation and soil microorganisms are intricately connected, with the former profoundly influencing the composition of the latter by altering abiotic factors, while the latter responds to vegetation by modifying soil physicochemical properties (*De Deyn & Van der Putten, 2005*; *Heerdt et al., 2017*). The diversity of microbial communities serves as a pivotal indicator of soil microbial characteristics (*Murphy et al., 2011*) and is a vital bioindicator for evaluating soil fertility (*Wei et al., 2018*). Consequently, it has emerged as a prominent area of research in plant-soil ecosystems in recent years.

To assess the variation in soil microbial community structure and diversity across different vegetation types, we conducted a study in Longzhou County, Chongzuo City, Guangxi Zhuang Autonomous Region. Specifically, we selected four distinct vegetation types: a woodland dominated by *Drypetes perreticulata*, a woodland dominated by *Horsfieldia hainanensis*, a maize (*Zea mays*) farmland, and a citrus (*Citrus reticulata*) field. Prior to human intervention, all of these sites were natural forests. Our objective was to compare the structure and diversity of soil bacterial and fungal communities among these vegetation types. To achieve this, we employed PCR amplification and high-throughput sequencing techniques. This allowed us to analyze and characterize the changes in soil microbial communities and their functions in response to different vegetation types.

## MATERIALS & METHODS

### Site information

The sites were situated in Longzhou County, Chongzuo City, Guangxi Zhuang Autonomous Region, China ($106°33'11''$–$107°12'43''$E, $22°8'54''$–$22°44'42''$N). It is characterized by a southern subtropical monsoon climate, characterized by high temperatures, abundant rainfall, and ample sunshine throughout the year. The region experiences a consistent pattern of hot and dry seasons, with approximately 350 frost-free days annually and a frost period lasting for 13 days. Four representative vegetation types were selected for the

experiment, namely a woodland with the dominant tree species *Horsfieldia hainanensis* (HH), a woodland with the dominant tree species *Drypetes perreticulata* (DP), a *Zea mays* farmland (ZM) and a *Citrus reticulata* farmland (CR). Both farmland areas were natural forests that were deforested to establish agricultural land. The *Zea mays* farmland has been under cultivation for 20 years, while the *Citrus reticulata* farmland has been cultivated for 10 years. Both farmland areas have been regularly fertilized as part of their daily management.

## Soil sampling

In December 2022, soil samples of HH, DP, ZM and CR were taken from 0 to 10 cm. According to the S-type sampling principle, eight soil cores of 0-10 cm were randomly selected in each plot and mixed to obtain a total of 16 samples. Each composite soil sample was carefully collected and placed in a plastic bag. The bag was labeled to ensure proper identification of the sample. Subsequently, the samples were transported to the laboratory in an ice box to maintain their integrity. In the lab, stones and plant residues such as roots and litter were removed from the soil. The soil material was then passed through two mm sieve to remove any coarse particles. Following this, the sieved soil was immediately transferred into 2 ml centrifuge tubes and frozen at a temperature of $-80\,°C$. This freezing process was carried out to facilitate the extraction of microorganisms from the soil at a later stage.

## Soil chemical properties determination

10g of soil sample was weighed and placed in a 50 mL conical flask, and double distilled water was added according to the principle of soil/water ratio of 2.5:1. After 2 min of high-speed shaking, the soil pH value was measured for half an hour by using a pH meter (Lei-ci PHS-3C). Dried soil samples of 10–13 mg were weighed and sealed in a tin container and then the total carbon and nitrogen contents of the soils were determined by the elemental analyzer (Elementar Vario EL III Germany). Using a molybdenum antimony anti-colorimetric method and $HClO_4$-$H_2SO_4$ heating digestion, the total phosphorus content of the soil was determined. 1 g of the sample was to be weighed in a cleaned and dried digestion tube. eight mL of concentrated sulfuric acid ($H_2SO_4$) were then required to be mixed, and allowed to soak for an overnight period. Finally, 10 drops of $HClO_4$ were supposed to be put in. The digestion tube might then be heated on the digestion machine until the boiling liquid is clarified. The sample was kept in a 100 mL volume bottle after digestion and cooling. The molybdenum-antimony resistance colorimetric method was used to perform colorimetric analysis on five mL of filtrate using a spectrophotometer (P4 UV-Visible China) and a 50 mL volumetric bottle (*Liu et al., 2022b*).

## DNA extraction and high-throughput sequencing

The OMEGA Soil DNA Kit (M5635-02) (Omega Bio-Tek, Norcross, GA, United States) was used to extract DNA from all the samples. A NanoDrop NC2000 spectrophotometer (Thermo Fisher Scientific, Waltham, MA, United States) was used to measure the quantity and quality of DNA and agarose gel electrophoresis for extracted DNA

(agarose concentration of 1.2%). With the assistance of the primers 338F (5′-ACTCCTACGGGAGGCAGCA-3′) and 806R (5′-GGACTACHVGGGTWTCTAAT-3′), the 16S V3_V4 region of the soil bacterium was amplified (*Claesson et al., 2009*). Additionally, the fungal ITS region was amplified using primers ITS5 (5′-GGAAGTAAAAGTCGTAACAAGG-3′) and ITS2 (5′-GCTGCGTTCTTCATCGATGC-3′) (*White et al., 1990*). The PCR system had a total volume of 25 µL, consisting of the following components: 5 µL of 5 × reaction buffer, 5 µL of 5 × GC buffer, 2 µL of dNTP (2.5 mM), 1 µL of forward primer (10 µM), 1 µL of reverse primer (10 µM), 2 µL of DNA template, 8.75 µL of ddH2O, and 0.25 µL of Q5 DNA polymerase. The amplification process involved an initial denaturation at 98 °C for 2 min, followed by cycling at 98 °C for 15 s, 55 °C for 30 s, and 72 °C for 30 s for a total of 25 cycles. A final extension was performed at 72 °C for 5 min, with a 10 °C hold for the 25 cycles. To purify the PCR amplicons, Vazyme VAHTSTM DNA Clean Beads from Vazyme in Nanjing, China were utilized. The Quant-iT PicoGreen dsDNA Assay Kit from Invitrogen in Carlsbad, CA, United States was employed for quantification. 250 pair-end sequencing was carried out using the Illlumina NovaSeq platform and NovaSeq 6000 SP Reagent Kit (500 cycles). Amplicons were pooled in equal amounts following the individual quantification step. The above operations were completed in Shanghai Personal Biotechnology Co., Ltd.

## Sequence analysis

Microbiome bioinformatics analysis was carried out using QIIME 2 (2019.4), with a slight modification based on the methodology described by *Bolyen et al. (2019)*. The initial step involved the removal of primers using the cutadapt plugin, following the demultiplexed of raw sequence data using the demux plugin (*Martin, 2011*). Subsequently, the DADA2 plugin was employed to perform quality filter, denoising, merging, and removal of chimera from the obtained sequences (*Callahan et al., 2016*). To construct a phylogenetic tree, non-singleton amplicon sequence variants (ASVs) were utilized in conjunction with the fasttree2 and mafft tools, which were implemented as part of the pipeline (*Katoh et al., 2002*; *Price, Dehal & Arkin, 2009*). Alpha-diversity metrics including Chao1 (*Chao, 1984*), *Shannon (1948)*, and Pielou's evenness (*Pielou, 1966*), were estimated by employing the diversity plugin. In addition, beta diversity metrics (Bray-Curtis dissimilarity) were calculated. The feature-classifier plugin was utilized to assign taxonomy to the ASVs using the naive Bayes classifier and two reference databases: SILVA Release 132 Database for bacteria and UNITE Release 8.0 Database for fungi, which were selected based on the studies conducted by *Kõljalg et al. (2013)* and *Bokulich et al. (2018)*, respectively.

## Data analytics

The main tools used to analyze sequence data were QIIME2 (2019.4) and R packages (v3.2.0; *R Core Team, 2015*) (New Zealand). Using the ASV table in QIIME2 (2019.4), alpha diversity indices at the ASV level, including the Chao1 richness index, Shannon index, and Pielou's evenness, were calculated and displayed as box plots. To visualize the differences in alpha diversity between the various specimen groups, the data in the table above were plotted as box plots using QIIME2 (2019.4) and the ggplot2 package for

the R package (v3.2.0; *R Core Team, 2015*). The significance of the differences could be confirmed using the Kruskal–Wallis rank sum test and the dunn'test as a post hoc test (*Wickham, 2016*). Based on the occurrence of ASVs across samples/groups regardless of their relative abundance, a Venn diagram was created to visualize the shared and unique ASVs among samples or groups using the R package "VennDiagram" (*Zaura et al., 2009*). PERMANOVA (Permutational Multivariate Analysis of Variance) was used to evaluate the significance of the differences in microbiota structure between groups (*Anderson & Willis, 2003*). Using abundance information from the top 20 orders of average abundance, the heatmap was produced using R's pheatmap package. A traditional multidimensional scaling (cMDScale) analysis technique is principal coordinates analysis(PCoA) (*Ramette, 2007*). This was accomplished by maximising the distance relationships between the original samples and expanding the sample distance matrix in low-dimensional space following projection. The number of permutation tests used in the "permanova" analysis of variance between groups was set to 999. The analysis of variance between groups was performed using the scikit-bio package in Python. After removing singletons from the feature list, the QIIME2 (2019.4) "qiime taxa barplot" was used to visualize the compositional distribution of each sample at the taxonomic levels of phylum and genus. In order to better understand the relationship between soil nutrient content and soil microbial community composition (phylum and genus level), we performed correlation heat map analysis of them based on Spearman rank correlation coefficient. This analysis was performed by the genescloud tools, A free online platform for data analysis (https://www.genescloud.cn) (*Liu et al., 2022a*). The differences between the chemical characteristics of the soils of the various vegetation types were examined using a one-way ANOVA test. Waller-Duncan post-hoc multiple comparisons were performed, and IBM SPSS Statistics 26 was used to process the data.

## RESULTS

### Comparative analysis of the chemical properties of soils of different vegetation types

The soil chemical properties of the four vegetation types were significantly different ($p < 0.01$; Table 1). pH value (mean 6.91), total soil carbon (mean 208.9 g/kg), total nitrogen(mean 13.69 g/kg), C/N (mean 15.27), N/P (mean 33.62) and C/P (mean 513.13) were all highest in DP and were significantly higher than in the other three vegetation types. The mean content of total phosphorus was 0.48 g/kg for ZM, which was significantly higher than the others ($p = 2.61 \times 10^{-3}$). HH and DP had significantly higher contents of each soil chemical property than CR and ZM.

### Comparative analysis of soil microbial community diversity in different vegetation types

In total, a total of 1,143,966 high-quality bacterial sequences and 1,040,630 high-quality fungal sequences were obtained from all samples, which provided an opportunity to delve deeper into the bacterial and fungal communities. On average, each sample contained 71,498 bacterial sequences and 65,039 fungal sequences. The number of bacterial sequences

**Table 1   Soil chemical properties of different vegetation types.**

|  | HH | DP | CR | ZM | F value | p value |
|---|---|---|---|---|---|---|
| **pH value** | 6.49 ± 0.11b | 6.91 ± 0.01a | 6.26 ± 0.01bc | 6.19 ± 0.13c | 15.029 | $2.29 \times 10^{-4}$ |
| **Total carbon/g kg$^{-1}$** | 70.22 ± 0.99b | 208.9 ± 3.45a | 20.6 ± 0.16c | 17.99 ± 0.21c | 2476.257 | $5.16 \times 10^{-17}$ |
| **Total nitrogen/g kg$^{-1}$** | 6.34 ± 0.06b | 13.69 ± 0.27a | 2.3 ± 0.01c | 1.81 ± 0.03d | 1544.172 | $8.71 \times 10^{-16}$ |
| **Total phosphorus/g kg$^{-1}$** | 0.35 ± 0.02b | 0.41 ± 0.02ab | 0.35 ± 0.01b | 0.48 ± 0.03a | 8.557 | $2.61 \times 10^{-3}$ |
| **C/N** | 11.08 ± 0.11b | 15.27 ± 0.11a | 8.96 ± 0.03d | 9.97 ± 0.04c | 1138.562 | $5.39 \times 10^{-15}$ |
| **N/P** | 18.39 ± 0.77b | 33.62 ± 1.87a | 6.54 ± 0.17c | 3.82 ± 0.23c | 177.404 | $3.34 \times 10^{-10}$ |
| **C/P** | 203.78 ± 8.95b | 513.13 ± 27.38a | 58.57 ± 1.63c | 38.05 ± 2.28c | 229.607 | $7.34 \times 10^{-11}$ |

**Notes.**

Data were average ± standard error.

Different lowercase letters meant significant difference at 0.05 level.

C/N, carbon to nitrogen ratio; N/P, nitrogen to phosphorus ratio; C/P, carbon to phosphorus ratio; HH, a woodland with the dominant tree species *Horsfieldia hainanensis*; DP, a woodland with the dominant tree species *Drypetes perreticulata*; ZM, a *Zea mays* farmland; CR, a *Citrus reticulata* farmland.

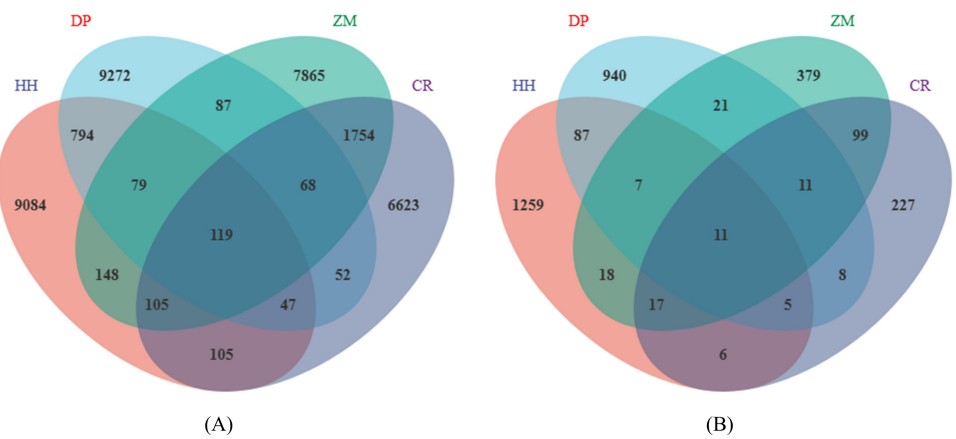

(A)                                                    (B)

**Figure 1   Venn diagram of soil microbial ASVs.** (A) Soil bacterial ASVs; (B) soil fungal ASVs; HH, a woodland with the dominant tree species *Horsfieldia hainanensis*; DP, a woodland with the dominant tree species *Drypetes perreticulata*; ZM, a *Zea mays* farmland; CR, a *Citrus reticulata* farmland.

ranged from 54,510 to 90,097 per sample, while the number of fungal sequences ranged from 51,968 to 77,522 per sample. All sequences were assigned to a total of 72,547 bacterial ASVs and 61,430 fungal ASVs. The number of soil bacterial ASVs in HH, DP, ZM, and CR was 10,481, 10,518, 10,226, 8,874, respectively (Fig. 1A). The number of ASVs in soil fungi was 1,410, 1,090, 563, 384 (Fig. 1B).

The number of shared soil bacterial ASVs among HH, DP, ZM, and CR was 119 (Fig. 1A). The 119 ASVs belonged to the phyla Acidobacteria, Actinobacteria, Chloroflexi, Firmicutes Gemmatimonadetes, Nitrospirae, Proteobacteria, Rokubacteria, Verrucomicrobia (Fig. 2A). Among them, Proteobacteria and Actinobacteria accounted for the highest percentage, with 45.39% and 33.61% respectively. Alphaproteobacteria was the dominant order shared by the four samples, with an ASV count of 46. The number of shared soil fungal ASVs among HH, DP, ZM, and CR was 11 (Fig. 1B). The ASVs common to all four samples were in the phyla Ascomycota (ASVs 81.82%) and Basidiomycota (ASVs 18.18%)

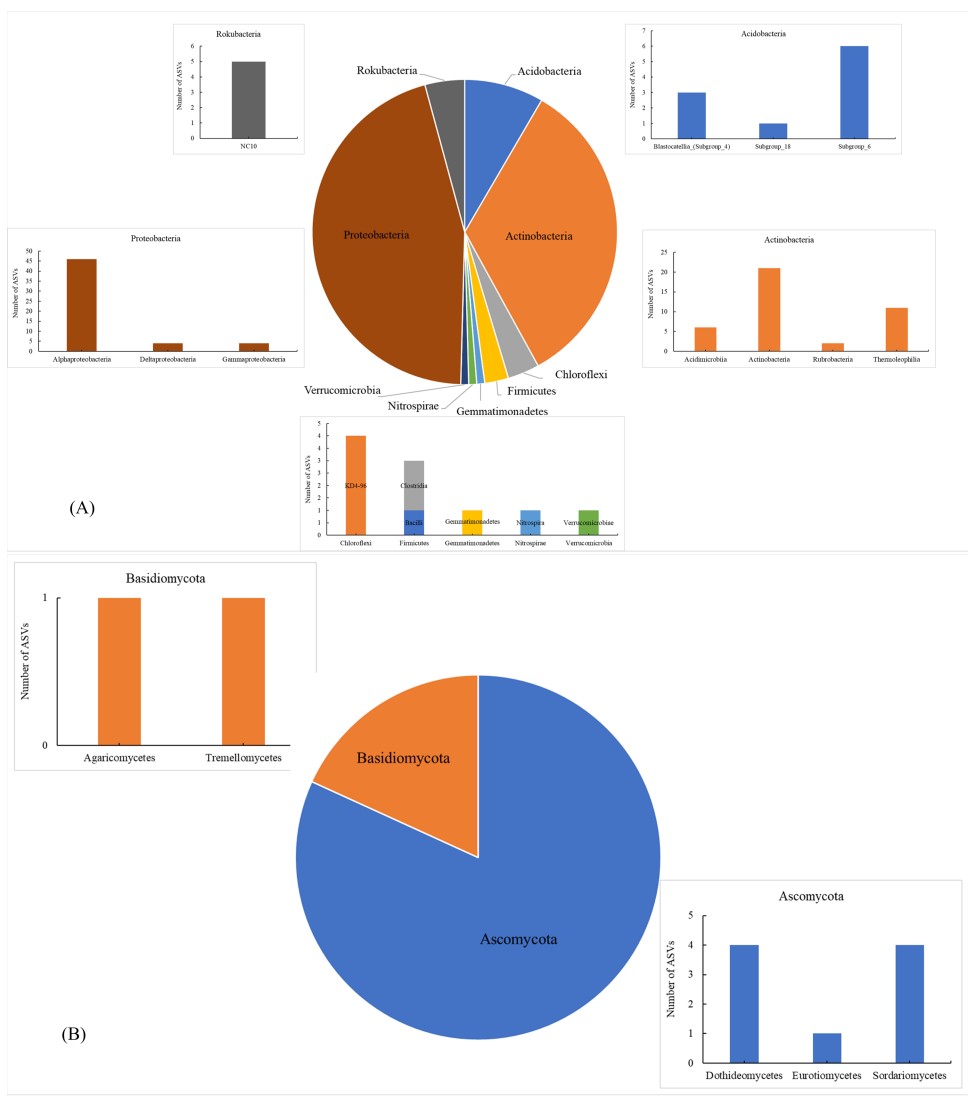

**Figure 2** **Soil microorganisms of different vegetation types share phyla and classes of microorganism.** (A) Bacteria; (B) fungi The horizontal coordinates were phyla, and those indicated in the legend were phyla. HH, a woodland with the dominant tree species *Horsfieldia hainanensis*; DP, a woodland with the dominant tree species *Drypetes perreticulata*; ZM, a *Zea mays* farmland; CR, a *Citrus reticulata* farmland.

(Fig. 2B). Dothideomycetes and Sordariomycetes were the dominant orders shared by the four samples.

The soil alpha diversity analysis of the different vegetation types was shown in Fig. 3. As a whole, the Chao 1 index, Shannon index and Pielou's evenness index were significantly different between the bacterial communities of the four soil samples (Fig. 3A). HH had the highest Chao 1 index (mean 4572.95), Shannon index (mean 10.88) and Pielou's evenness index (mean 0.91), while the lowest mean Chao 1 index, Shannon index and Pielou's evenness index were found in CR at 3628.18, 10.22 and 0.88, respectively. Of the four soil samples, only HH and CR differed significantly between the two pairs (Chao 1 index $p =$

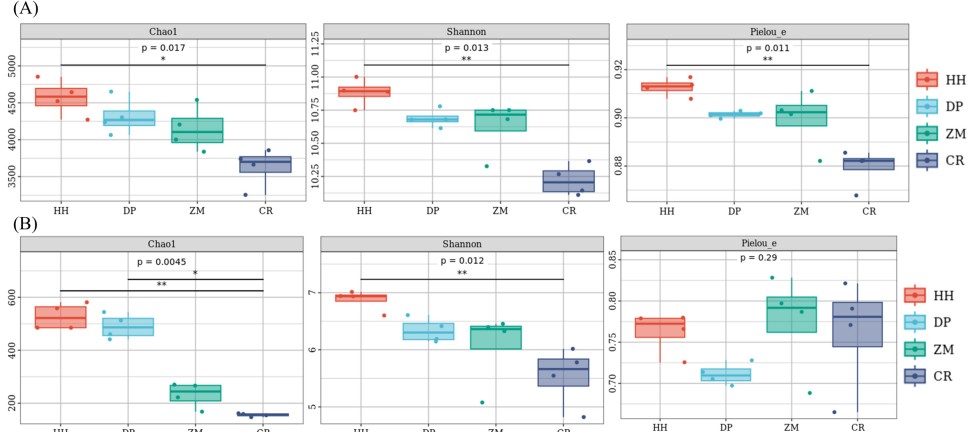

**Figure 3** **Box plot of alpha diversity of soil microbial communities of different vegetation types.** (A) Bacterium, (B) fungus. An asterisk (*) indicates p value < 0.05; two asterisks (**) indicate p value < 0.01; HH, a woodland with the dominant tree species *Horsfieldia hainanensis*; DP, a woodland with the dominant tree species *Drypetes perreticulata*; ZM, a *Zea mays* farmland; CR, a *Citrus reticulata* farmland.

0.014; Shannon index $p = 0.0065$ and Pielou index $p = 0.005$; Table S1). The rest of the treatments were not significantly different between the two pairs (Table S1). Unlike the bacterial community, only the Chao 1 index and Shannon index were significantly different between the fungal communities of the four soil samples, while the Pielou's evenness index was not significantly different between them (Fig. 3B). The mean values of the Chao 1 index and Shannon index followed the same pattern as the bacterial community, being HH (527.34; 6.87) >DP (489.56; 6.34) >ZM (231.67; 6.06) >CR (155.71; 5.54). The Chao 1 index and Shannon index of HH were significantly higher than that of CR ($p = 0.0065$; Table S1). The Chao 1 index of CR was significantly lower than that of DP ($p = 0.038$; Table S1). In contrast to the bacterial community, ZM had the highest mean Pielou's evenness index and DP the lowest. the mean Pielou's evenness index of HH was only 0.0004 higher than that of CR. Soil bacterial communities had higher Chao 1 index, Shannon index, and Pielou's evenness index than soil fungal communities.

## Comparative analysis of soil microbial community composition in different vegetation types

The relative abundance of soil microbial (bacterial and fungal) communities in each of the four vegetation types was analyzed at the phylum level and genus level, and communities with relative abundances greater than 1% were selected for comparative analysis (Table S2). Eight bacterial phyla with relative abundance greater than 1% were Actinobacteria, Proteobacteria, Acidobacteria, Chloroflexi, Gemmatimonadetes, Rokubacteria, Planctomycetes, Bacteroidetes (Fig. 4A). Among the four vegetation types, the phylum Actinobacteria, Proteobacteria, Acidobacteria, and Chloroflexi had average relative abundances greater than 10%, and they were also among the top 5 phyla in terms of the number of ASVs shared by all soil bacterial communities (Fig. 2A, Table S2). HH had the highest relative abundance of Proteobacteria and Acidobacteria of all samples at 36.31%

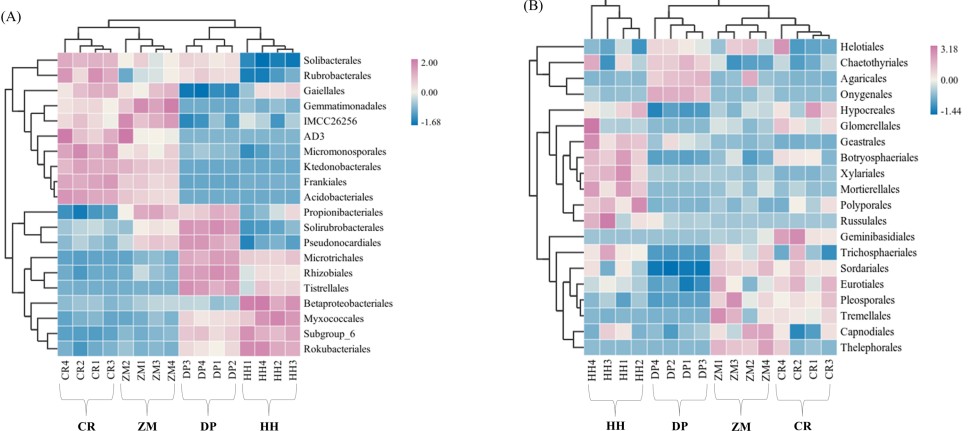

**Figure 4** **Heatmap and cluster analysis based on relative abundance of the top 20 orders identified in soil microbial communities.** (A) Bacterium; (B) fungus. The samples are grouped according to their similarity to each other. In the figure, brown represents the genus with lower abundance in the corresponding sample, green represents the genus with higher abundance, and the color change represents the level of abundance. HH, a woodland with the dominant tree species *Horsfieldia hainanensis*; DP, a woodland with the dominant tree species *Drypetes perreticulata*; ZM, a *Zea mays* farmland; CR, a *Citrus reticulata* farmland.

and 15.38%, while Actinobacteria had the lowest relative abundance at 24.81%. CR had the highest relative abundance of Actinobacteria (42.99%) and Chloroflexi (18.57%) and the lowest of Proteobacteria (17.39%). Acidobacteria had the lowest relative abundance of 9.63% in ZM, while Chloroflexi had the lowest relative abundance of 3.28% in DP, the lowest of all.

At the fungal phylum level, relative abundances greater than 1% were Ascomycota, Basidiomycota, Mortierellomycota (Table S2). The only soil fungal phyla with average relative abundances greater than 10% were Ascomycota and Basidiomycota, which were also the phyla to which all the sampled fungal communities shared ASVs (Fig. 2B, Table S2). The relative abundance of the remaining eight phyla, with the exception of Mortierellomycota, was not higher than 0.01%. CR had the highest relative abundance of Ascomycota (85.46%) and the lowest relative abundance of Basidiomycota (12.44%). However, in DP, Ascomycota had the lowest relative abundance at 55.17% and Basidiomycota had the highest relative abundance at 31.91%.

At the level of soil bacterial genera in the four vegetation types, CR had not only the highest relative abundance of *AD3* (7.62%), *Acidothermus* (7.70%), *Rubrobacter* (2.16%) and *JG30-KF-AS9* (2.87%), but also the lowest relative abundance of *Subgroup_6* (1.97%), *67-14* (1.42%), *Rokubacteriales* (1.40%), *bacteriap25* (0.65%), *Gaiella* (1.61%), *Solirubrobacter* (0.87%). HH had the highest relative abundance of *Subgroup_6* (8.92%), *Rokubacteriales* (4.44%), *bacteriap25* (4.64%), *Gaiella* (1.90%) and the lowest relative abundance of *AD3* (0.001%), *Rubrobacter* (0.82%) and *JG30-KF-AS9* (0.01%) had the lowest relative abundance. The highest relative abundance of DP's *67-14* (7.25%) and *Solirubrobacter* (2.30%) was found among them, while *Acidothermus* had the lowest

relative abundance of 0.007% (Table S2). At the soil fungal genus level, the top ten genera in terms of CR relative abundance totaled 54.18%, the highest of them all, with ZM in second place at only 27.01% and HH the lowest at 21.11% (Table S2). *Fusarium* had the highest relative abundance in CR(17.84%) and the lowest relative abundance in DP (1.997%). *Aspergillus* had the highest relative abundance in HH (5.89%) and the lowest in DP (3.34%). *Hygrocybe* had a relative abundance of 17.03% in DP but was not distributed in HH and ZM. *Paramyrothecium* was not contained in DP and ZM and *Acrophialophora* was not distributed in HH and DP.

Based on the soil microbial community order level, the four sample sites were clustered using the average algorithm. At the bacterial order level, the HH and DP communities were combined into a single branch, and the ZM and CR communities were in the same branch (Fig. 4A). The bacterial communities of the two sample sites in the same branch, *i.e.,* between HH and DP and between ZM and CR, were more similar. At the fungal order level, the ZM and CR communities merged into one branch and then coalesced with DP; the HH community was a separate branch (Fig. 4B). This indicated that the CR and ZM communities were more similar and least similar to HH. This clustering result reflected that the soil microbial community characteristics of the four sample site communities were closely related to the sample site plant community composition.

Further, a PERMANOVA test was conducted to examine the differences in soil microbial community composition between the four vegetation types (Table 2). As a result, the differences in soil microbial community composition between the four samples reached a significant level ($p < 0.05$). Among them, DP and ZM had the most significant differences in microbial community composition. The study showed that the type of above-ground vegetation was an important driver of structural differences in soil microbial community composition, and that differences in soil microbial community composition were more pronounced between vegetation types.

## Relationship between soil nutrient factors and microbial community composition

The relative abundance of microphyla, with the exception of Basidiomycota, showed significant correlations with soil nutrient factors when their relative abundances exceeded 1% (Fig. 5). Specifically, the relative abundances of Actinobacteria, Chloroflexi, Gemmatimonadetes, and Ascomycota were significantly negatively correlated with soil pH ($r = -0.55$, $p < 0.05$; $r = -0.68$, $p < 0.01$; $r = -0.72$, $p < 0.01$; $r = -0.52$, $p < 0.05$), total carbon ($r = -0.68$, $p < 0.01$; $r = -0.89$, $p < 0.01$; $r = -0.75$, $p < 0.01$; $r = -0.75$, $p < 0.01$), total nitrogen ($r = -0.68$, $p < 0.01$; $r = -0.89$, $p < 0.01$; $r = -0.74$, $p < 0.01$; $r = -0.71$, $p < 0.01$), C/N ($r = -0.65$, $p < 0.01$; $r = -0.77$, $p < 0.01$; $r = -0.94$, $p < 0.01$; $r = -0.62$, $p < 0.05$), N/P ($r = -0.66$, $p < 0.01$; $r = -0.77$, $p < 0.01$; $r = -0.94$, $p < 0.01$; $r = -0.62$, $p < 0.01$), and C/P ($r = -0.66$, $p < 0.01$; $r = -0.91$, $p < 0.01$; $r = -0.77$, $p < 0.01$; $r = -0.71$, $p < 0.01$). On the other hand, the relative abundances of Acidobacteria, Rokubacteria, and Planctomycetes were significantly positively correlated with soil pH ($r = 0.53$, $p < 0.05$; $r = 0.50$, $p < 0.05$; $r = 0.63$, $p < 0.01$), total carbon ($r = 0.64$, $p < 0.01$; $r = 0.73$, $p < 0.01$; $r = 0.71$, $p < 0.01$), total nitrogen ($r = 0.63$, $p < 0.01$; $r = 0.72$,

**Table 2 Analysis of differences between groups.**

| Group1 | Group2 | Permutations | pseudo-F | *p*-value | *q*-value |
|--------|--------|--------------|----------|-----------|-----------|
| | | **Bacterium** | | | |
| HH | DP | 999 | 27.24 | 0.030 | 0.037 |
| HH | ZM | 999 | 41.07 | 0.031 | 0.037 |
| HH | CR | 999 | 63.42 | 0.024 | 0.037 |
| DP | ZM | 999 | 44.83 | 0.023 | 0.037 |
| DP | CR | 999 | 82.10 | 0.037 | 0.037 |
| ZM | CR | 999 | 5.50 | 0.035 | 0.037 |
| | | **Fungus** | | | |
| HH | DP | 999 | 11.89 | 0.033 | 0.033 |
| HH | ZM | 999 | 6.90 | 0.025 | 0.033 |
| HH | CR | 999 | 7.10 | 0.032 | 0.033 |
| DP | ZM | 999 | 13.85 | 0.021 | 0.033 |
| DP | CR | 999 | 15.12 | 0.027 | 0.033 |
| ZM | CR | 999 | 3.89 | 0.022 | 0.033 |

Notes.

HH, a woodland with the dominant tree species *Horsfieldia hainanensis*; DP, a woodland with the dominant tree species *Drypetes perreticulata*; ZM, a *Zea mays* farmland; CR, a *Citrus reticulata* farmland.

$p < 0.01$; $r = 0.71$, $p < 0.01$), C/N ($r = 0.83$, $p < 0.01$; $r = 0.62$, $p < 0.05$; $r = 0.68$, $p < 0.01$), N/P ($r = 0.63$, $p < 0.01$; $r = 0.77$, $p < 0.01$; $r = 0.72$, $p < 0.01$), and C/P ($r = 0.64$, $p < 0.01$; $r = 0.76$, $p < 0.01$; $r = 0.70$, $p < 0.01$). Furthermore, the relative abundances of Proteobacteria and Bacteroidetes were significantly positively correlated with soil total carbon ($r = 0.73$, $p < 0.01$; $r = 0.78$, $p < 0.01$), total nitrogen ($r = 0.73$, $p < 0.01$; $r = 0.78$, $p < 0.01$), C/N ($r = 0.56$, $p < 0.05$; $r = 0.57$, $p < 0.05$), N/P ($r = 0.73$, $p < 0.01$; $r = 0.79$, $p < 0.01$), and C/P ($r = 0.73$, $p < 0.01$; $r = 0.79$, $p < 0.01$), while showing significant negative correlations with soil total phosphorus ($r = -0.52$, $p < 0.05$; $r = -0.61$, $p < 0.05$). Additionally, the relative abundance of Rokubacteria and Mortierellomycota were negatively correlated with soil total phosphorus content ($r = -0.55$, $p < 0.05$; $r = -0.57$, $p < 0.05$). Notably, there was no significant correlation observed between the relative abundance of Basidiomycota and soil nutrient content ($p > 0.05$). Among these microphyla, Gemmatimonadetes showed the strongest negative correlation with soil pH ($r = -0.72$) and C/N ($r = -0.94$), Chloroflexi had the highest negative correlation with soil total carbon ($r = -0.89$), total nitrogen ($r = -0.89$), N/P ($r = -0.92$), and C/P ($r = -0.91$), and Bacteroidetes had the highest negative correlation with soil total phosphorus content ($r = -0.61$).

At the genus level, our analysis revealed significant associations between soil nutrient factors and bacterial genera, with relative abundance exceeding 1% (Fig. 6A). Soil pH exhibited significant correlations with the relative abundance of *AD3*, *Acidothermus*, *Rokubacteriales*, *Gaiella*, *JG30-KF-AS9*, and *IMCC26256* ($p < 0.05$). Among these genera, only *Rokubacteriales* showed a positive correlation ($r > 0$). Notably, the relative abundances of *Subgroup_6*, *67-14*, *Rokubacteriales*, *bacteriap25*, and *Solirubrobacter* displayed significant positive correlations with total carbon, total nitrogen, N/P, and C/P ratios ($r > 0$, $p < 0.05$).

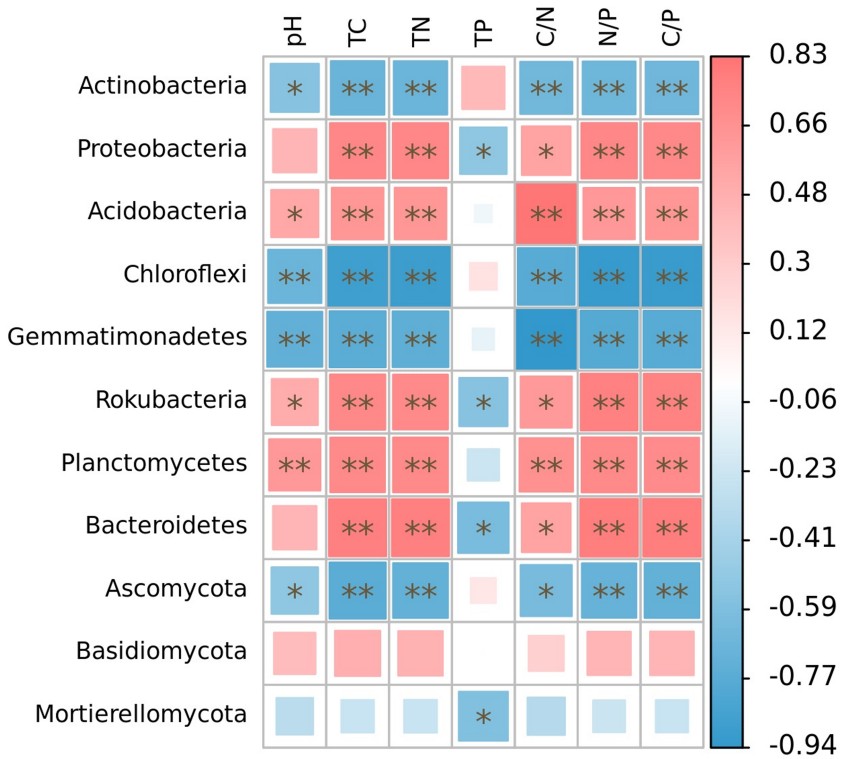

**Figure 5** Correlation heatmap of correlations between phyla with relative abundance greater than 1% and soil nutrient factors based on Spearman's algorithm. HH, a woodland with the dominant tree species *Horsfieldia hainanensis*; DP, a woodland with the dominant tree species *Drypetes perreticulata*; ZM, a *Zea mays* farmland; CR, a *Citrus reticulata* farmland.

Conversely, these nutrient factors exhibited negative correlations with the abundance of *AD3*, *Acidothermus*, *JG30-KF-AS9*, *IMCC26256*, and *TK10* ($r < 0$, $p < 0.05$). Additionally, soil C/N ratio demonstrated significant positive correlations with the relative abundance of *Subgroup_6*, *Rokubacteriales*, and *bacteriap25* ($r > 0$, $p < 0.05$), while exhibiting negative correlations with the abundance of *AD3*, *Acidothermus*, *JG30-KF-AS9*, *IMCC26256*, and *TK10* ($r < 0$, $p < 0.05$). Furthermore, only *Subgroup_6*, *Rokubacteriales*, and *Gaiella* displayed negative correlations with soil total phosphorus content ($r < 0$, $p < 0.05$), whereas the relative abundances of *Rubrobacter*, *TK10*, and *RB41* exhibited positive correlations with soil total phosphorus content ($r > 0$, $p < 0.05$). Notably, the relative abundances of *Mycobacterium* and *Bradyrhizobium* did not exhibit any significant correlations with the examined soil nutrient factors ($p > 0.05$). The relative abundances of Fusarium, *Talaromyces*, *Basidioscus*, *Acrophialophora* and *Dokmaia* exhibited negative correlations with the other factors, with the exception of soil total phosphorus ($r < 0$, $p < 0.05$). Conversely, a significant positive correlation was observed with the relative abundance of *Hygrocybe* ($r > 0$, $p < 0.05$) (Fig. 6B). The relative abundances of *Penicillium*, *Humicola* and *Saitozyma* were negatively correlated with soil total carbon, total nitrogen, N/P and C/P ($r < 0$, $p < 0.05$). Additionally, the relative abundance of *Saitozyma* displayed a negative

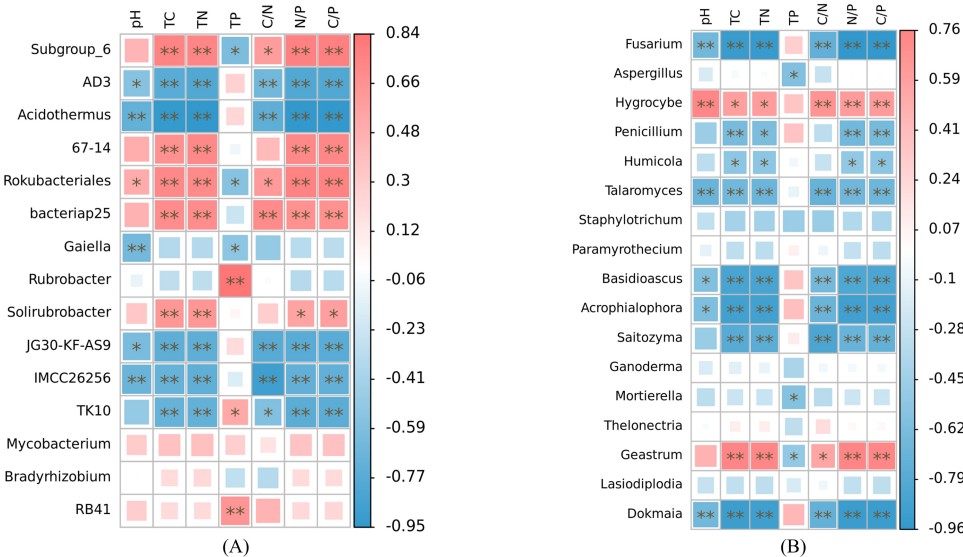

**Figure 6** **Correlation heat maps of soil nutrient factors in bacteria (A) and fungi (B) with relative abundance greater than 1% based on Spearman algorithm.** TC, total carbon; TN, total nitrogen; TP, total phosphorus; C/N, carbon to nitrogen ratio; N/P, nitrogen to phosphorus ratio; C/P, carbon to phosphorus ratio.

correlation with soil C/N ($r < 0$, $p < 0.01$). The relative abundances of *Aspergillus* and *Mortierella* were negatively correlated solely with soil total phosphorus content ($r < 0$, $p < 0.05$). On the other hand, the relative abundance of *Geastrum* exhibited significant positive correlations with soil total carbon, total nitrogen, C/N, N/P and C/P ($r > 0$, $p < 0.05$), while displaying a negative correlation with soil total phosphorus content ($r < 0$, $p < 0.05$).

## DISCUSSION

Soil is widely recognized as a crucial habitat for microbial communities, making it a key component of the Earth's ecosystem (*De Vrieze, 2015*). In agriculture and forestry, soil microbial communities, composed of bacteria and fungi, play a vital role in the cycling of materials within the ecosystem (*Štursová et al., 2012*). The activities of soil microorganisms are intricately linked environment, and alterations in environmental factors, such as human disturbances and variations in vegetation types, can have a significant impact on microbial composition and cause changes in their distribution patterns (*Nakamura et al., 2003*). It has been observed that topsoil from agricultural land (ZM and CR) possessed a lower number of ASVs compared to woodland areas (HH and DP). Futhermore, microbial community diversity, richness and evenness were lower in farmland than in woodland. This trend may be attributed to the frequent tillage practices in farmland, which disturb the soil and subsequently reduce microbial community richness (*Zhang et al., 2018*). Additionally, the alpha diversity index of soil microbial communities in *Zea may* is higher than that of *Citrus reticulata*, despite both being agricultural crops. This difference could be explained
by the fact that *Zea may* is an annual herb, while Citrus reticulata is a perennial tree. Repeated tillage activities in *Zea may* cultivation may moderately disturb the microbial community, while the addition of a substantial amount of litter during the wilt period of *Zea may* could enhance microbial community richness (*Bressan et al., 2008*; *Araujo et al., 2023*). The significant influence of different vegetation types on soil microbial community diversity can also be attributed to variations in plant characteristics and rooting systems. Perennial plants, with their extensive root systems and greater carbon and nitrogen availability through root deposition and turnover, provide a favorable environment for microbial colonization and nutrient cycling. In contrast, annual crops like maize have a shorter photosynthetic life cycle, whereas citrus trees typically have shallow rooting depths. Consequently, these factors contribute to lower soil microbial community diversity (*Liang et al., 2012*; *Zhang et al., 2017*).

The microbial communities in soil exhibit variations in structure and diversity across different vegetation types, as highlighted by studies conducted by *Szoboszlay et al. (2017)*, *George et al. (2019)*, and *Santos et al. (2020)*. Our study also revealed significant disparities in soil microbial composition between woodland and agricultural land, with the highest similarity observed between ZM and CR soil microbial communities. The dominant bacterial phyla in the topsoil of the different vegetation types were found to be the consistent and included Actinobacteria, Proteobacteria, Acidobacteria, and Chloroflexi. The prevalence of these phyla has been previously demonstrated in multiple studies (*Peiffer et al., 2013*; *Byers et al., 2020*; *Gao et al., 2020*), and demonstrates the prevalence and importance of these phyla. *Acidothermus* is an important genus of bacteria in the Acidobacteria. The relative abundance of *Acidothermus* was higher in agricultural soils than in woodlands due to the fact that *Acidothermus* could decompose difficult components and convert them into organic components of the soil after breaking them down into humus, while growing crops would increase the number of soil inter-root bacteria, so the bacterial population showed a higher number in agricultural fields than in woodlands (*Rajkumar et al., 2010*). Fertilizer application also resulted in significant nutrient inputs, which could also increase the abundance and activity of *Acidothermus*, which was essential for the soil carbon and nitrogen cycle (*Kielak et al., 2016*). Furthermore, the abundance of Gemmatimonadetes was observed to be higher in ZM and CR compared to other land use types. Members of this phylum play a role in essential nutrient recycling and the decomposition of cellulose and lignin, highlighting their significance in ecosystem functioning (*Xu et al., 2019*).

In terms of soil fungal community composition, Ascomycota emerged as the predominant phylum across different vegetation types, exhibiting the highest relative abundance. Basidiomycota was the next most prevalent phylum, which aligns with findings from previous studies (*Porras-Alfaro et al., 2011*; *Maestre et al., 2015*; *Prober et al., 2015*). Moreover, our study revealed that the total abundance of fungal genera within the top 10 CR relative abundances at the genus level was considerably higher than in the other three soils, with agricultural soils exhibiting a higher overall abundance compared to woodland soils. This is can be attributed to the influence of human activities, particularly agricultural management, which has a discernible impact on factors such as vegetation composition,

soil water and temperature levels, and mineralization of soil organic matter. These factors subsequently lead to structural changes in soil fungal communities, resulting in variations in diversity and the emergence of new species (*Arévalo-Gardini et al., 2020*). At the genus level, *Fusarium* was relatively abundant in ZM and CR soils. This can be attributed to the fact that certain Fusarium species are major causal agents of Fusarium crown rot and Fusarium root rot in crops (*Beccari, Covarelli & Nicholson, 2011*).

The various vegetation types exhibited differences not only in their role of regulating the microclimate of the habitat, but also in their processes of material cycling. These processes involved the input of plant-derived nutrients, as well as the decomposition, transformation and accumulation of nutrients. These factors, in turn, influenced the collaborative evolution of the physico-chemical properties of the soil system and the microbial community (*Zhang et al., 2013*; *Tian et al., 2017*; *Delgado-Baquerizo et al., 2018*). The present study found a significant correlation between the relative abundances of dominant genus taxa in the soil microbial community and environmental factors. Furthermore, there were significant differences in the relationships between different taxa and various environmental factors. These findings may be attributed to the varying degrees of influence that different environmental factors have on microorganisms in the soil, thereby closely relating to the ecological niche of microorganisms (*Wan & He, 2020*). Previous studies have reported significant variations in soil physicochemical factors among different vegetation communities (*Xu et al., 2014*; *Chen et al., 2018*). Moreover, research has shown that abiotic properties of the soil, such as soil total nitrogen and total carbon content, are the primary regulators of the structure and composition of soil bacteria and fungi, (*Thakur & Geisen, 2019*). In this study, the relevant heat map analysis also confirmed this point. There were significant differences in the relative abundance of soil microorganisms (bacteria and fungi) with different soil nutrient factors.

## CONCLUSIONS

In conclusion, soil microorganisms play a crucial role in forest ecosystems by facilitating interactions between plants and soil. The composition and distribution of soil microorganisms are influenced by different vegetation types, making it essential to investigate the structural and functional diversity of soil microbial communities to understand plant-soil microbial-soil relationships and the underlying driving mechanisms. In this study, we employed 16S rRNA and ITS sequencing to analyze the characteristics of soil microbial communities in four distinct vegetation types in the subtropical region of Guangxi, China, yielding noteworthy findings. We observed significant variations in soil microbial structure and diversity among the different vegetation types, with the most pronounced differences occurring between woodland and agricultural land. Overall, woodland soils exhibited greater soil microbial community diversity compared to agricultural soils. The dominant phylum among soil fungi across all samples was Ascomycota while the dominant phylum varied between Proteobacteria in woodland soils and Actinobacteria in agricultural soils among bacteria. Moreover, we identified a significant correlation between soil nutrient content of different vegetation types and the

relative abundance of soil microorganisms at both the phylum and genus levels. Future research should consider incorporating complementary soil microbial analysis techniques to further elucidate the characteristics and disparities of microbial communities across different vegetation types.

### Funding
This work was supported by the National Natural Science Foundation of China (No. 32271843). The funders had no role in study design, data collection and analysis, decision to publish, or preparation of the manuscript.

### Grant Disclosures
The following grant information was disclosed by the authors:
National Natural Science Foundation of China: 32271843.

### Competing Interests
The authors declare there are no competing interests.

### Author Contributions
- Yong Jiang conceived and designed the experiments, analyzed the data, prepared figures and/or tables, and approved the final draft.
- Wenxu Zhu performed the experiments, prepared figures and/or tables, and approved the final draft.
- Keye Zhu analyzed the data, prepared figures and/or tables, and approved the final draft.
- Yang Ge analyzed the data, prepared figures and/or tables, and approved the final draft.
- Wuzheng Li performed the experiments, authored or reviewed drafts of the article, and approved the final draft.
- Nanyan Liao conceived and designed the experiments, authored or reviewed drafts of the article, and approved the final draft.

### Data Availability
The soil bacterial communities sequence reads are available at PRJNA939272 and the fungal communities sequence reads are available at PRJNA939403.

### Supplemental Information
Supplemental information for this article can be found online at http://dx.doi.org/10.7717/peerj.16260#supplemental-information.

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
