# Peer review of "Similarities and differences in the microbial structure of surface soils of different vegetation types"

_PeerJ, doi:10.7717/peerj.16260_

## Round 0.1 · original submission · Major Revisions

Please revise the manuscript thoroughly based on the reviewers' reports. When submitting your revised manuscript, please make sure that a point-by-point response letter will be also attached. Thank you for submitting your work to PeerJ for consideration. We look forward to receiving your revised manuscript.

**Language Note:** The review process has identified that the English language must be improved. PeerJ can provide language editing services - please contact us at copyediting@peerj.com for pricing (be sure to provide your manuscript number and title). Alternatively, you should make your own arrangements to improve the language quality and provide details in your response letter. – PeerJ Staff

Reviewer 1 ·

Basic reporting

Comments
Dear editors:
The manuscript “Similarities and differences in the microbial structure of surface soils of different vegetation types” has several flaws that need to be revised:

In this study, the surface soil microbial structure of different vegetation types was studied in detail, which has certain significance for the study of microbial structure and function. There are still some major problems in the depth of presentation, experimental method, statistical analysis, graph drawing and result description.

1.Line 127 random plots are not scientific. Field sampling methods should be chosen, such as S-type sampling or five-point sampling.
2. The description of Soil chemical properties determination in the materials method is too concise in general, which only explains the method without elaborating the process.
3. Line 156-157, using the Illlumina NovaSeq platform and NovaSeq 6000 SP Reagent Kit (500 cycles)。The tenses and grammar of sentences need to be modified.
4. Does the number of bacterial ASVs specific to the four forest soils described in lines 215-217 not correspond to the figure?
5. The name description of a microorganism or species in line 221 should be in italics.
6. The color scheme of the article graphics is too monotonous; please read the most cutting-edge articles and choose the most appropriate color scheme.
7. There are still grammatical errors and logical confusion in the language expression of the article, which needs to be polished by fluent English speakers.
8. Your workload is large, but the graphic information is too little, only introducing the door and can not reflect the actual problem, because the gap of the door is very subtle, the top several bacteria are almost the same, now need to add the genera, re-write the article and the overall layout.
9. Microbial network analysis and correlation heat map analysis are missing in this paper.

Experimental design

1.Line 127 random plots are not scientific. Field sampling methods should be chosen, such as S-type sampling or five-point sampling.
2. The description of Soil chemical properties determination in the materials method is too concise in general, which only explains the method without elaborating the process.
3. Line 156-157, using the Illlumina NovaSeq platform and NovaSeq 6000 SP Reagent Kit (500 cycles)。The tenses and grammar of sentences need to be modified.

Validity of the findings

In this study, the surface soil microbial structure of different vegetation types was studied in detail, which has certain significance for the study of microbial structure and function.

Additional comments

There are still some major problems in the depth of presentation, experimental method, statistical analysis, graph drawing and result description.

Annotated reviews are not available for download in order to protect the identity of reviewers who chose to remain anonymous.

Reviewer 2 ·

Basic reporting

The English language should be improved to ensure that an international audience can clearly understand your manuscript. Some examples where the language could be improved include lines 32, 33, 395, etc. Some sentences are too long. I suggest you have a colleague who is proficient in English and familiar with the subject matter review your manuscript, or contact a professional editing service.

Experimental design

Lack of soil physical property data, such as SWC, EC, pH.
The discussion section needs further revision.

Validity of the findings

no comment

Additional comments

1. Line 119-121: How long have HH and DP been planted.
2. Line 125: Is it the 9 point sampling method? What is the sampling depth when sampling?
3. Line 134: Soil physical data such as SWC, EC and pH were lacking in the manuscript.
4. Line 334: The discussion needs to be rewritten, and many explanations are not relevant to the study data and are not discussed in conjunction with the findings.
5. Line 335-338: This sentence does not have much to do with the article. It is recommended to revise and rewrite it.
6. Line 346-347: Your study did not measure the soil aggregates.
7. Line 348-351: Could this be caused by additional nitrogen and phosphorus additions?
8. Line 351-352: This statement is imprecise, ecological niche is not reflected in your study.
9. Line 353: This model explains plant communities, not the same as microbial communities, and this explanation is a bit of a stretch.
10. Line 362: There is no root biomass data in your study, please discuss it in conjunction with your own research results.
11. Line 377-379: The amount of fertilizer applied is not described in the methods section.
12. Line 394: Missing bracket
13. Line 395-399: Most importantly, your study has no data on the physical properties of the soil, and hopefully you can add to it.
14. Line 397: Root secretions have been mentioned many times in introduction and discussion. This is only one aspect. Differences in plant litter can also cause changes in microbial community structure and biogeochemical cycles.
15. Line 407-409: I'm not sure what this statement is meant to convey, and it's not discussed combined with the results.

---

## Round 0.2 · accepted · Accept

The authors have addressed the reviewers' comments and concerns. Therefore, I would like to endorse the publication of the manuscript.

Reviewer 2 ·

Basic reporting

no comment

Experimental design

no comment

Validity of the findings

no comment

Additional comments

no comment